# Prevalence and risk factors of developing cardiac arrhythmia in patients presenting to the emergency department with electrical injuries

Ramiz Yazıcı[1]*, Bensu Bulut[2], Murat Genç[3], Medine Akkan Öz[2], Serkan Şahin[4], Ayşenur Gür[5], Hüseyin Mutlu[6]

1 Department of Emergency Medicine, Istanbul Kanuni Sultan Suleyman Training and Research Hospital, Health Science University, Istanbul, Turkey, 2 Department of Emergency Medicine, Ankara Gulhane Training and Research Hospital, Ankara, Turkey, 3 Department of Emergency Medicine, Ankara Training and Research Hospital, University of Health Sciences, Ankara, Turkey, 4 Department of Emergency Medicine, Ankara Bilkent City Hospital, University of Health Sciences, Ankara, Turkey, 5 Department of Emergency Medicine, Etimesgut Şehit Sait Ertürk State Hospital, Ankara, Turkey, 6 Department of Emergency Medicine, Aksaray Training and Research Hospital, Aksaray University, Aksaray, Turkey

* dr.ramiz.yazici@gmail.com

## Abstract

### Background

Cardiac arrhythmias following electrical shocks are significant concerns in emergency medicine, yet predictive factors remain unclear.

### Objective

This study aimed to investigate the prevalence and risk factors of cardiac arrhythmias in patients presenting with electrical injuries to the emergency department.

### Methods

In this retrospective study conducted between January 2019 and December 2023, we analysed 189 patients aged ≥18 years who presented with electrical injuries. Patients were divided into two groups based on whether or not they developed an arrhythmia. Demographics, clinical characteristics, and laboratory parameters were compared between groups.

### Results

Cardiac arrhythmia developed in 21.2% (n = 40) of patients. The arrhythmia group showed significantly higher mean age (32.4 ± 16.7 vs 26.5 ± 14.8 years, p = 0.023) and high-voltage exposure rates (≥1000 V) (p = 0.015). Multivariate analysis identified age (OR: 1.02, 95% CI: 1.01–1.05), CK > 850 U/L (OR: 1.32, 95% CI: 1.17–1.81), troponin > 250 ng/mL (OR: 1.23, 95% CI: 1.09–1.72), lactate > 2.1 mmol/L (OR: 2.51, 95% CI: 1.67–5.91), and high voltage (OR: 2.03, 95% CI: 1.64–5.39) as independent risk

**Data availability statement:** All relevant data are within the manuscript and its Supporting Information files.

**Funding:** The author(s) received no specific funding for this work.

**Competing interests:** The authors have declared that no competing interests exist.

factors. ROC analysis showed high voltage (AUC: 0.804) as the strongest predictor of developing arrhythmia.

## Conclusion

This study demonstrates that high voltage exposure, advanced age, and elevated biomarkers are significant predictors of developing arrhythmia in patients with electrical injuries. These findings may guide clinical decision-making regarding cardiac monitoring in the emergency department.

## 1. Introduction

Electrical injuries, ranging from low-voltage household accidents to high-voltage industrial accidents, are a major public health problem affecting the whole world and can cause serious morbidity and mortality [1,2]. While the incidence of electric burns in Europe is 4–5%, in the United States of America, approximately 4400 cases of electrical shock are reported annually, 3–8% of which result in death [2–4]. Cardiac complications, especially arrhythmias are one of the most important causes of mortality and morbidity in electrical injuries [3–5].

Trauma caused by electrical injuries can lead to multiple organ damage by affecting the skin, musculoskeletal system, cardiovascular system and nervous system [3–5]. However, electric currents passing through the body tends to follow the least resistance; i.e., the neurovascular structures. Therefore, cardiac involvement is common in electrical injuries and sometimes severe myocardial damage can occur even without significant skin damage [5]. Cardiac effects of electric current are caused by direct thermal damage, electrical simulation and electroporation mechanisms [6,7]. These effects can result in various arrhythmias both in early and late stages [7]. Because of the risk of cardiac injury, both prehospital and hospital emergency room personnel must be trained and alert [8]. For the detection of these arrhythmias, close cardiac monitoring is required in emergency departments, but there is no consensus in the literature regarding which patients require cardiac monitoring following electrical injuries. Various factors affecting the decision for cardiac monitoring in electrical injuries have been identified. These include the voltage level, the path the current takes through the body, the duration of exposure and the patient's co-morbidities [9]. Although the risk of cardiac complications increases in high-voltage injuries, serious complications can be seen in low-voltage injuries too [10]. Despite these factors, while some authors advise 24-hour cardiac monitoring for all patients with electrical injuries, others consider it necessary for only high risk patients [7,11]. These differences in approach make it difficult to strike a balance between the use of limited hospital resources and patient safety.

In this study, we aimed to investigate the prevalence of cardiac arrhythmias and the factors associated with the development of arrhythmia in patients presenting to the emergency department (ED) after an electric shock, in order to help ED physicians assess these patients and take the necessary precautions. In this way,

emergency physicians and pre-hospital medical care providers will be able to identify patients at risk of developing cardiac arrhythmias at an early stage and will be informed about the need for closer and monitored observation.

## 2. Materials and methods

### 2.1. Study design and participants

This retrospective study was conducted at the Adult Emergency Department of Bilkent City Hospital, a tertiary care hospital with an average of 65,000 patients per month, and pre-hospital emergency medical services with an average of 40,000 cases per month between 1 January 2019 and 31 December 2023. Ethical approval from the Bilkent City Hospital Ethics Committee was obtained prior to the study (Number: TABED 2-24-326 and Date: 26.06.2024). The study population consisted of patients over the age of 18 years who were brought in to the emergency department of our hospital by pre-hospital emergency medical services due to an electrical accident. Patients were excluded if they had a history of known rhythm disorders, coronary artery disease, were younger than 18 years of age, were pregnant, did not have a routine blood test results or electrocardiogram (ECG), or if their records were inaccessible, patients who present to the hospital on their own initiative. The study was conducted in accordance with the tenets of the Declaration of Helsinki. Informed consent was not required due to the retrospective nature of the study. The dataset used for this study was fully anonymized before analysis, and none of the authors had access to identifying personal information at any stage. The data were accessed for research purposes on 5 July 2024, after obtaining ethics committee approval on 26 June 2024.

### 2.2. Study protocol and data collection

All patients presenting with electrical accidents were evaluated in the Emergency Department's Red Zone and standard treatment protocols (haemodynamic support, fluids, analgesic support) were implemented. For patients who presented with cardiac arrest, in-hospital basic and advanced life support and post-CPR care were provided in accordance with AHA guidelines (11). Demographics, clinical characteristics, admission status, ECGs and laboratory data of these patients were obtained from the hospital's digital database. Electrical accidents with 1000 volts and above were classified as high voltage events [12]. Patients were divided into two groups according to whether or not they developed a cardiac arrhythmia. Arrhythmias were classified as non-specific S-T and T segment changes, right bundle branch block, sinus tachycardia (>100/min) or bradycardia (<50/min), ventricular premature beats, supraventricular extrasystoles, atrial fibrillation, ventricular fibrillation, ventricular tachycardia and pulseless tachycardia. Patients who developed arrhythmias during follow-up were first evaluated by the emergency physician, and those requiring further assessment were subsequently consulted to cardiology. For patients who received a cardiology consultation, the electrocardiograms (ECGs) were interpreted by a cardiology specialist. In the absence of a cardiology consultation, the ECGs were evaluated based on the interpretation made by the emergency physician responsible for the patient's care. For patients without any documented ECG interpretation, the ECGs were reviewed by two emergency medicine specialists participating in the study, each with at least ten years of clinical experience (H.M. and R.Y.).

### 2.3. Statistical analysis

Study data were analysed using IBM SPSS 27.0 software (Armonk, NY: IBM Corp.). Study variables were reported as frequency, percentage, mean, standard deviation, median, mode and IQR. The chi-square test ($\chi^2$) was used to compare qualitative data. The distribution of the data was assessed using the Kolmogorov-Smirnow test, skewness and kurtosis, and graphical methods (histogram, Q-Q plot, StemandLeaf, boxplot). The study used Independent Samples t-test (t-test in independent groups) for data that is normally distributed and Mann-Whitney U-test for data that is not normally distributed. Logistic regression (binary logistic regression) test was used to determine the risk ratios of the variables, and receiver operating characteristic (ROC) analysis was performed to calculate the sensitivity and specificity and to determine the optimal cut-off values. The statistical significance level was accepted as $\alpha = 0.05$.

## 3. Results

A total of 189 patients who met the criteria were included in the study. The study design is shown in Fig 1. The mean age of patients included in the study was 27.9±15.3 years and 87% were male. 21.2% (40) of the patients developed arrhythmias. The demographic and clinical characteristics and laboratory parameters of the patients according to whether or not they developed an arrhythmia are shown in Table 1. The most common rhythms in patients who developed arrhythmia were sinus tachycardia and non-specific S-T and T changes; rhythm distribution in patients who developed arrhythmia is shown in Fig 2.

Age, CK, troponin, lactate and high voltage were statistically significantly higher in the arrhythmia group than in the non-arrhythmia group (p:0.023, p:0.008, p:0.046, p<0.001, p=0.012 respectively). Statistically significant parameters were included in the regression model. In multivariate regression analysis, age (OR: 1.02, 95%CI: 1.01–1.05), CK>850 U/L (OR: 1.32, 95%CI: 1.17–1.81), troponin >250ng/ml (OR: 01.23, 95%CI: 1.09–1.72), lactate (OR: 2. 51, 95%CI: 1.67–5.91) and industrial current (OR: 2.03, 95%CI: 1.64–5.39) were identified as independent risk factors (Table 2). We performed ROC analysis to determine the predictive power of cardiac arrhythmia in electrical accidents. We found that high voltage was a predictor of arrhythmia at a cut-off voltage of ≥1000 volts (AUC: 0.804) and was more valuable than lactate (AUC: 0.782) and CK (AUC: 0.792) (Table 3).

## 4. Discussion

Electrical shocks are a major cause of morbidity and mortality worldwide [11,13,14]. Although it affects all systems in the body, its effect on the cardiovascular system is the most lethal one [11,15]. Cardiovascular effects can include myocardial

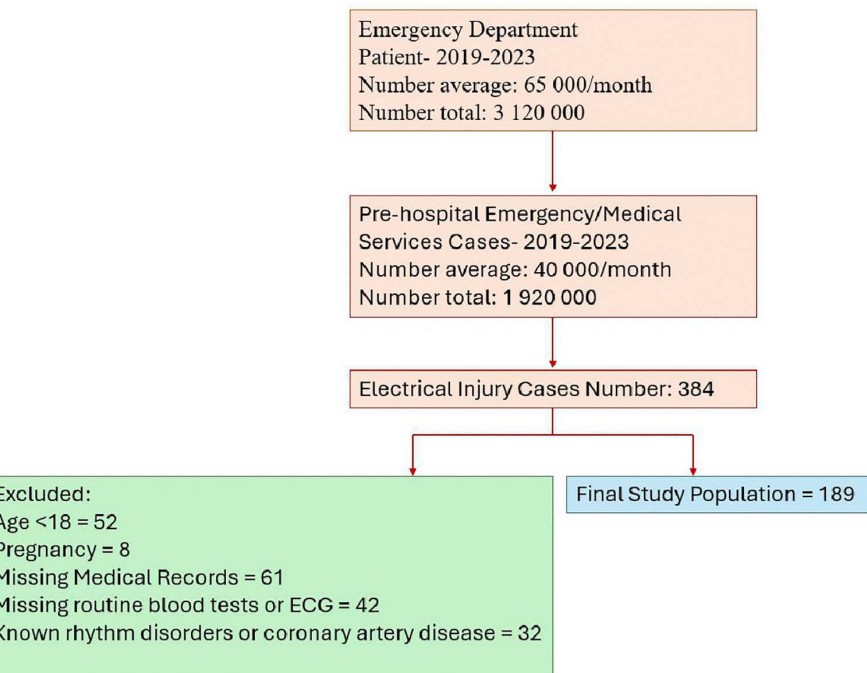

**Fig 1. Flow diagram of study population selection.** This flowchart outlines the selection process for the final study population evaluating electrical injuries between 2019 and 2023. From an initial cohort of approximately 3,120,000 emergency department visits and 1,920,000 pre-hospital emergency medical service cases, a total of 384 electrical injury cases were identified. Following exclusion of patients due to age under 18 (n=52), pregnancy (n=8), missing medical records (n=61), absence of routine blood tests or ECG (n=42), and known rhythm disorders or coronary artery disease (n=32), the final study population consisted of 189 eligible cases.

**Table 1. Demographic and clinical characteristics of patients with electrical injury.**

| Characteristic | All patients (n = 189) | Cardiac Arrhythmias | | P val |
| --- | --- | --- | --- | --- |
| | | Yes (n:40) | No (n:149) | |
| **Age (years)**, mean ± SD | 27.9±15.3 | 32.4±16.7 | 26.5±14.8 | **0.023** |
| **Gender**, n (%) | | | | |
| Male | 135 (87.0) | 33 | 102 | 0,254 |
| Female | 54 (13.0) | 7 | 47 | 0,312 |
| **Type of Electrical Current**, n (%) | | | | |
| Low voltage < 1000v | 97 (51.1) | 4 | 93 | |
| High voltage ≥1000v | 92 (48.9) | 36 | 56 | **0,015** |
| **Location**, n (%) | | | | |
| Urban | 157 (89.3) | 31 | 126 | 0,105 |
| Rural | 32 (10.7) | 9 | 23 | 0.213 |
| **Reaction Time of Pre-hospital Medical Services Coordination Center (min)** | 9,2±13,7 | 9,9±16,7 | 9,1±11,7 | 0.085 |
| **Reaction Time of Pre-hospital Medical Services Holding Station (s)** | 13,9±29,5 | 12,8±19,5 | 14,2±31,5 | 0.091 |
| **Time of Presentation**, n (%) | | | | |
| 00:00-07:59 | 19 (8.4) | 6 | 13 | 0.064 |
| 08:00-15:59 | 104 (48.9) | 27 | 77 | 0.103 |
| 16:00-23:59 | 66 (42.7) | 7 | 59 | 0.12 |
| **Initial Laboratory Values**, mean ± SD | | | | |
| Urea | 25,0 (16,0 − 46,0) | 25,0 (18,0 − 45,0) | 24,0 (16,0 − 46,0) | 0,787 |
| Creatinine | 1,0 (0,8 −1,5) | 1,1 (0,9 −1,6) | 1,0 (0,7 −1,5) | 0,928 |
| Lactate, mmol/L | 2.1(0.7-4.5) | 4.1 (1.6-6.7) | 1.4 (0.9-2.1) | **< 0.001** |
| CK (U/L) | 1751.5±5733.2 | 2845.6±6124.3 | 1415.8±5598.7 | **0.008** |
| Troponin (ng/mL) | 225.0±1515 | 415.0±1823 | 208.3±1315 | **0.013** |
| Potassium (mmol/L) | 4.1±0.4 | 4.2±0.3 | 4.0±0.4 | 0.084 |
| **Co-morbidity** | | | | |
| DM | 5 | 3 | 2 | 0.351 |
| HTN | 2 | 1 | 1 | 0.56 |
| Asthma | 4 | 2 | 2 | 0.65 |
| Other Co-morbidities | 5 | 1 | 4 | 0.52 |
| No Co-morbidities | 173 | 20 | 153 | 0.96 |
| **Trauma context; n (%)** | | | | |
| Work related | 142 | 35 | 107 | 0.049 |
| Non-work related | 44 | 3 | 41 | |
| Suicide attempt | 3 | 2 | 1 | 0.14 |
| **Duration of Hospitalisation (days)** | 6,2±12,6 | 8,3±11,6 | 5,1±10,6 | 0.015 |
| **Patient Outcome** | | | | |
| Hospitalisation | 46 | 31 | 15 | 0.01 |
| Discharge from ED | 135 | 1 | 134 | |
| Death | 8 | 8 | 0 | |

DM: diabetes mellitus; HTN: hypertension; CK: serum creatine kinase.

damage, arrhythmias and even cardiac arrest [13,15]. It has been suggested that electric currents may cause permanent damage to the cardiac conduction tissue and thus cause predisposition to early and late stage arrhythmias [11]. Cardiac arrhythmia, which can occur in the early stages after an electrical shock and be life-threatening [2,4], is a major concern

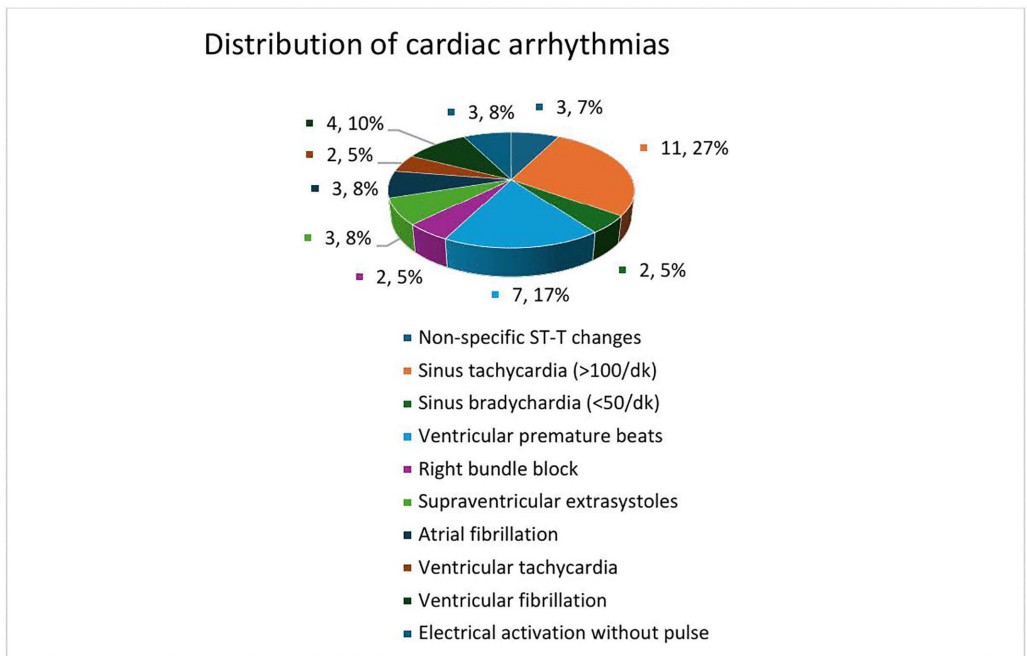

**Fig 2. Distribution of cardiac arrhythmias in patients with electrical injuries.** This pie chart illustrates the distribution of various cardiac arrhythmias observed in patients presenting with electrical injuries. The most common arrhythmia was sinus tachycardia (>100 bpm), observed in 27% of cases, followed by non-specific ST-T changes (17%) and ventricular premature beats (10%). Other identified arrhythmias included sinus bradycardia (<50 bpm), right bundle branch block, supraventricular extrasystoles, atrial fibrillation, ventricular tachycardia, ventricular fibrillation, and electrical activation without pulse. Each category is represented as a percentage of the total arrhythmic events identified during the study.

**Table 2. Univariate and multivariate analysis of predictive factors for arrhythmia in electric shocks.**

| Variables | Univariate logistic regression | | Multivariate logistic regression | |
|---|---|---|---|---|
| | OR (95% CI) | P-value | OR (95% CI) | P-value |
| Age (per year) | 1.03 (1.01–1.05) | 0.023* | 1.02 (1.01–1.05) | 0.032* |
| CK > 850 U/L | 1.42 (1.18–1.71) | 0.008* | 1.32 (1.17 – 1.81) | 0.023* |
| Troponin >250 ng/mL | 1.31 (1.09- 1.62) | 0.046* | 1.23 (1.09– 1.72) | 0.084 |
| Lactate >2,1 mmol/L | 2.31 (1.44–3.71) | <0.001 | 2.51 (1.64–5.91) | 0.001* |
| High voltage ≥1000 | 1.89 (1.22–2.93) | 0.012* | 2.03 (1.64-5.39) | 0.021* |

OR: odds ratio; CI: confidence interval; CK: serum creatine kinase; v: Volt; *p-value < 0.05.

for emergency physicians. Therefore, we planned to evaluate the risk of cardiac arrhythmia in patients following an electrical shock and to investigate the associated factors. In our study, we found that cardiac arrhythmia was not detected in 21.2% of patients following an electrical shock, and the risk of developing arrhythmia was increased significantly in patients with high voltage exposure (especially work related injuries), advanced age, high CK, troponin and lactate levels.

In our study, we found that sinus tachycardia and non-specific ST-T changes were the most common rhythm disorders in patients who developed arrhythmia. Waldmann et al. reported that tachycardia due to the activation of the sympathetic nervous system was common in electrical injuries [16]. A detailed study by Jensen et al. showed that electric current increases the risk of arrhythmia by causing direct myocardial damage and repolarisation abnormalities [2]. Our study highlighted that ventricular arrhythmias were more common with high voltage electrical injuries. In their study analysing

**Table 3. Analysis of the area under the ROC curve for arrhythmia in electrical accidents.**

| Parameter | AUC | Cut-Off |
|---|---|---|
| Age, per year | 0.654 | 45 years |
| CK, U/L | 0.782 | 850 U/L |
| Troponin, ng/mL | 0.551 | 250 ng/mL |
| Lactate, mmol/L | 0.792 | 2 mmol/L |
| Industrial Current | 0.804 | ≥1000 V |

ROC: receiver operating characteristic; AUC: area under the curve; CK: serum creatine kinase, V: Volt.

the natural course of electrical injuries, Solem et al. reported that ST-T changes and tachyarrhythmias were predominant in the early stages [17].

CK and troponin levels are important markers used to assess myocardial damage after an electrical shock [18]. Electric currents can cause damage to skeletal muscles and myocardium, leading to increased CK levels [16]. While it has been emphasised that especially CK-MB results should be interpreted with caution because of peripheral muscle damage [18], troponin is a marker specific to myocardial damage, and the damage and necrosis in myocardial cells caused by electrical shock may lead to elevated troponin levels [19]. The study by Douillet et al. shows that troponin levels are used to predict myocardial damage due to electrical shock [20]. Purdue et al. reported that elevated cardiac enzymes are an important risk factor for developing arrhythmia in electrical injuries [18]. In our study, a significant relationship has been found between elevated troponin and CK-MB levels and the development of arrhythmia. The risk of arrhythmia was significantly increased in patients with troponin >250 ng/mL (OR: 1.23, 95% CI: 1.09–1.72).

We also found lactate levels to be an important predictor of arrhythmia development after electrical shock (OR: 2.51, 95% CI: 1.67–5.91). The risk of arrhythmia was significantly increased in patients with lactate >2.1 mmol/L. Dahl et al. reported in their experimental study that electric currents caused direct myocardial damage and that this damage could be detected by biochemical markers [7]. In light of these findings, it can be said that elevated lactate levels are an important risk factor that requires close cardiac monitoring, especially in high-voltage electrical injuries.

The relationship between the level of voltage and the development of arrhythmia is one of the most important findings of our study. The rate of arrhythmia development was found to be significantly higher in the high voltage group (≥1000V) (AUC: 0.804). In a series of 65 cases, DiVincenti et al. emphasised that cardiac complications were more frequent in patients with high voltage exposure and that these patients required close follow-up [21]. Wilbourn et al. also reported that voltage level was one of the most important factors in determining the clinical course of electrical injuries [22].

There was no association between whether patients came from urban or rural areas and the development of arrhythmias. In addition, there was no delay in the transport times of pre-hospital medical services, and this factor was also not related to the occurrence of arrhythmias (p-values > 0.05). In the study by Drahap et al., no significant association was found between pre-hospital time and the mortality of trauma patients [23].

Our study has several limitations. Firstly, there were difficulties in accessing some data due to its retrospective design. This limited the assessment of late stage follow-up outcomes in particular. In addition, our study population consists of patients presenting to a single centre and therefore the generalisability of our findings may be limited. However, our study also has its strengths. Few studies in the literature have analysed the risk factors associated with developing arrhythmia following electrical shocks in such a comprehensive manner. Both the sufficient number of patients and the availability of detailed cardiac monitoring data increase the reliability of our study.

## 5. Conclusions

In conclusion, determining the factors that predict the development of arrhythmias following electrical shocks is important in terms of efficient use of resources and patient safety. As found in our study, closer follow-up of patients with high-voltage exposure, CK > 850 U/L, troponin >250 ng/mL, lactate >2.1 mmol/L may allow early diagnosis and treatment of possible cardiac complications. This approach will ensure that high-risk patients are not overlooked. However, future studies could look more closely at the factors associated with the development of arrhythmias after electrical shocks by analysing larger groups of patients and different types of electrical shocks. In addition, strategies and treatments to prevent the development of arrhythmias following electrical shocks can be explored.

## Supporting information

**S1 Text. STROBE-checklist-electricalInjury.**
(PDF)

## Acknowledgments

There is no person, instution or company to acknowledgement.

**Animal and Human Rights Statement:** All procedures performed in studies involving human participants were in accordance with the ethical standards of the institutional and/or national research committee and with the 1964 Helsinki Declaration and its later amendments or comparable ethical standards. No animal studies were carried out by the authors for this article.

**Scientific Responsibility Statement:** All authors declare that they are responsible for the scientific content of this article. This responsibility covers all aspects of the manuscript including the study design, data collection, analysis and interpretation, writing, preparation, scientific review, and approval of the final version of the article.

## Author contributions

**Conceptualization:** Bensu Bulut, Murat Genç, Hüseyin Mutlu.

**Data curation:** Bensu Bulut.

**Formal analysis:** Hüseyin Mutlu.

**Methodology:** Bensu Bulut, Murat Genç, Medine Akkan Öz, Serkan Şahin.

**Project administration:** Bensu Bulut.

**Writing – original draft:** Bensu Bulut, Murat Genç, Medine Akkan Öz, Ayşenur Gür.

**Writing – review & editing:** Ramiz Yazici, Bensu Bulut.

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
