## [Decision Letter · Decision Letter 0]

30 Oct 2025

Dear Dr. bulut,

Thank you for submitting your manuscript to PLOS ONE. After careful consideration, we feel that it has merit but does not fully meet PLOS ONE’s publication criteria as it currently stands. Therefore, we invite you to submit a revised version of the manuscript that addresses the points raised during the review process.

We look forward to receiving your revised manuscript.

Kind regards,

Ahmet Çağlar, Associate Professor

Academic Editor

PLOS ONE

Journal Requirements:

3. Please include captions for your Supporting Information files at the end of your manuscript, and update any in-text citations to match accordingly. Please see our Supporting Information guidelines for more information: http://journals.plos.org/plosone/s/supporting-information .

Reviewers' comments:

Reviewer's Responses to Questions

**Comments to the Author**

1. Is the manuscript technically sound, and do the data support the conclusions?

Reviewer #1: Yes

Reviewer #2: Yes

2. Has the statistical analysis been performed appropriately and rigorously?

Reviewer #1: Yes

Reviewer #2: Yes

3. Have the authors made all data underlying the findings in their manuscript fully available?

Reviewer #1: No

Reviewer #2: Yes

4. Is the manuscript presented in an intelligible fashion and written in standard English?

Reviewer #1: Yes

Reviewer #2: Yes

Reviewer #1: General Evaluation:

Your study focuses on the factors influencing the development of arrhythmias in patients with electrical injuries who are transported to the emergency department by prehospital medical services. Identifying these factors is valuable for patient monitoring and preparedness for early intervention. The topic is relevant and has the potential to contribute meaningfully to the literature. I would like to congratulate you on addressing this important issue.

Abstract:

The abstract is generally clear and appropriately reflects the overall content of the study.

Introduction:

The importance of the topic is well emphasized. However, the existing gap in the literature should be stated more explicitly. The originality of the study should be articulated clearly in one or two sentences.

Materials and Methods:

Ethics committee approval is mentioned in the text, but the approval number and date are not provided. Please include this information.

You have indicated that the patients were transported by prehospital medical services. However, outpatient admissions are not listed among your exclusion criteria. Please add them to the exclusion criteria.

Results:

The resolution of the graph in Figure 1 is low. The visual materials should be revised in higher resolution.

Discussion:

The findings are interpreted appropriately; however, in the comparison with the literature, please also address and compare any of your remaining results that were not discussed.

Conclusion:

You argue that your findings may help to “prevent unnecessary hospital admissions” but this statement is somewhat overstated. The study is primarily relevant to the medical community and is not aimed at patient education. This statement should therefore be removed.

Language:

The English used throughout the manuscript is accurate and sufficiently clear.

Best Regars.

Reviewer #2: Dear Author,

I would like to share some comments that I believe may be helpful for improving your manuscript:

Abstract: The abstract is well-written, clear, and appropriately focused on the study’s objectives. It is satisfactory in its current form.

Introduction: I believe you should elaborate further on how your study contributes to the existing literature and what specific gap it aims to address.

Materials and Methods: The name of the ethics committee and the date of ethical approval should be added. Please clarify the rationale for defining an electric voltage above 500 as “high.” It would be appropriate to cite a reference supporting this threshold.

Who evaluated the ECGs, and were the detected arrhythmias consulted with the cardiology department? Please specify.

Results: You categorized comorbidities as DM, HTN, and asthma. Were there no other comorbidities? If none were present, please indicate this explicitly in Table 1.

Discussion: In the discussion section, there is no comparison with the literature regarding some variables presented in Table 1 (e.g., time to admission). Please include such comparisons or consider removing these variables from the table.

References: Please check whether your references are up to date.

Tables: All tables should follow a consistent format. Please revise accordingly.

Figures: Figure 1 is not clearly readable; its quality should be improved.

Language: The manuscript does not require language editing. The writing is clear and sufficiently fluent.

Sincerely.

**Do you want your identity to be public for this peer review?** For information about this choice, including consent withdrawal, please see our Privacy Policy

Reviewer #1: No

Reviewer #2: No

---

## [Author Response · Author response to Decision Letter 1]

12 Nov 2025

Thank you for your valuable contributions. Our responses to the reviewers’ comments and the details of the revisions made can be found in the "Responses to Reviewers" file. Sincerely.

---

## [Editor Report · Decision Letter 1]

14 Nov 2025

Prevalence and Risk Factors of Developing Cardiac Arrhythmia in Patients Presenting to the Emergency Department with Electrical Injuries

PONE-D-25-51435R1

Dear Dr. Yazici,

We’re pleased to inform you that your manuscript has been judged scientifically suitable for publication and will be formally accepted for publication once it meets all outstanding technical requirements.

Kind regards,

Ahmet Çağlar, Associate Professor

Academic Editor

PLOS ONE
---

## [Editor Report · Acceptance letter]

PONE-D-25-51435R1

PLOS ONE

Dear Dr. Yazici,

I'm pleased to inform you that your manuscript has been deemed suitable for publication in PLOS ONE. Congratulations! Your manuscript is now being handed over to our production team.

Kind regards,

on behalf of

Dr. Ahmet Çağlar

Academic Editor

PLOS ONE